# The association of COVID-19 employment shocks with suicide and safety net use: An early-stage investigation

**Michihito Ando**[1]*, **Masato Furuichi**[2]

1 Department of Economics, Rikkyo Univeristy, Tokyo, Japan, 2 Department of Economics, Teikyo Univeristy, Tokyo, Japan

☯ These authors contributed equally to this work.

* michihito.ando@rikkyo.ac.jp

**Data Availability Statement:** All the data used in the paper will be available at the GitHub repository of Michihito Ando: https://github.com/michihito-ando.

## Abstract

This paper examines whether the COVID-19-induced employment shocks are associated with increases in suicides and safety net use in the second and third quarters of 2020. We exploit plausibly exogenous regional variation in the magnitude of the employment shocks in Japan and adopt a difference-in-differences research design to examine and control for possible confounders. Our preferred point estimates suggest that a one-percentage-point increase in the unemployment rate in the second quarter of 2020 is associated with, approximately, an additional 0.52 suicides, 28 unemployment benefit recipients, 88 recipients of a temporary loan program, and 10 recipients of public assistance per 100,000 population per month. A simple calculation based on these estimates suggests that if a region experienced a one-percentage-point increase in the unemployment rate caused by the COVID-19 crisis in the second quarter of 2020, which is roughly equivalent to the third-highest regional employment shock, this would be associated with 37.4%, 60.5%, and 26.5% increases in the total, female, and male suicide rates respectively in July 2020 compared with July 2019. These results are primarily correlational rather than causal due to the limitation of our data and research design, but our baseline findings are robust to several different model specifications.

## 1 Introduction

The COVID-19 pandemic has caused serious infections and deaths around the world, and it is becoming clearer that its economic and social consequences are also tremendous. The COVID-19 pandemic and subsequent social-distancing policies have caused a sharp contraction of economic and social activities. National governments around the world have been trying to mitigate socio-economic damages by introducing new emergency cash benefits and expanding existing safety net programs. There are daily news reports of job loss, poverty, mental disorders, and even suicides that have been directly or indirectly induced by the COVID-19

**Funding:** M.A. currently receives research grants from JSPS (Japan Society For the Promotion of Science) KAKENHI and he has used one of the research grants for this study. The grant number is JP20K01733. The website of JSPS (Japan Society For the Promotion of Science) is as follows: https://www.jsps.go.jp/english/index.html No party, including JSPS, had the right to review this manuscript prior to its circulation.

**Competing interests:** The authors have declared that no competing interests exist.

crisis. Numerous social-science studies of the impact of the COVID-19 crisis have also been published on a daily basis.

However, it is still not well understood how the COVID-19 economic shocks, such as a sharp increase in unemployment, have affected society. Previous social science studies have tended to focus on how the COVID-19 crisis *as a whole*, including social distancing policies, has affected economic and social outcomes, but fewer studies have directly examined how COVID-19-induced *employment shocks* are associated with social outcomes such as mental and financial distress.

In this paper, we provide an early-stage analysis on how the COVID-19-induced employment shocks are associated with suicide and financial distress during and after the first wave of COVID-19 in Japan. We exploit an increase in the regional unemployment rate as an indicator of the economic shocks because a deteriorating employment environment can directly and indirectly harm the well-being of workers and their families. We focus on suicide and safety net use as social outcomes because the former can be interpreted as a devastating consequence of deteriorated well-being and the latter are administrative indices that reflect the number of people or households that suffer from distressed living and financial conditions.

As a research design, we exploit a considerable regional variation in the magnitude of the COVID-19-induced increase in unemployment: regional (i.e., prefecture-level) employment shocks were larger in metropolitan areas and regions with popular sightseeing spots where a greater portion of employed people work in the service industry. We then adopt a difference-in-differences (DID) research design with an event-study specification and examine how the regional employment shocks caused by COVID-19 are associated with the regional trends of suicide rates and safety net participation before and during the crisis. As for safety net programs, we examine five programs in Japan's three-tier safety net system addressing unemployment and poverty: unemployment benefits (first tier), two temporary loans and a temporary housing benefit (second tier), and a public assistance program (third tier).

Our study is primarily descriptive and correlational and we do not argue that our research design perfectly solves the problem of estimation bias. On the one hand, we implement DID estimations that incorporate pre-determined covariates in order to control for possible confounding COVID-19 impacts through other pathways than COVID-19-induced employment shocks. In addition, we also implement several robustness checks using different model specifications. On the other hand, we nonetheless carefully interpret our estimation results as correlations between the COVID-19-induced employment shock and the outcomes of interest because of the possibility that some unobserved COVID-19-related factors are still not controlled for.

Our preferred point estimate suggests that a one-percentage-point increase in the unemployment rate in the second quarter (i.e., April to June) of 2020 is associated with an additional 0.522 suicides per 100,000 population in the subsequent month, July 2020. The size of this estimate is not negligible given that the suicide rate was 1.395 in July 2019 (based on the estimated days of death), implying a 37.4% increase in the suicide rate. The counterpart estimates and the rates of increase for female and male suicide rates are 0.518 (60.5%) and 0.521 (26.5%) respectively.

The counterpart point estimates for the safety net programs show that the same one-percentage-point increase in the regional unemployment rate is associated with an additional 27.9 unemployment benefit recipients, an additional 87.8 temporary loan program recipients, and an additional 9.7 public assistance recipients per 100,000 population in July 2020. These results suggest that all three tiers of safety net programs responded to COVID-19-induced employment shocks and the second-tier safety net of temporary loans in particular played an important role in this period.

The contributions of our paper are three-fold. First, we provide early-stage plausible evidence of association between the COVID-19-induced employment shocks and mental and financial distress such as suicide and safety net use. This association has rarely been studied so far, although several epidemiological and economic studies investigate suicide [1–7], suicide ideation and mental health outcomes [8–13], and socio-economic outcomes including unemployment and safety net use [14–18] under the COVID-19 pandemic. There is also research that examines relationship between unemployment and safety net use under the COVID-19 crisis based on a social survey [19] and some studies also point out that the rise in unemployment caused by the COVID-19 pandemic is expected to lead to an increase in the suicide rate [20, 21]. Another study examines the possible long-run impacts of the economic consequences of COVID-19 on various socio-economic outcomes from a historical perspective [22]. But few studies directly investigate how COVID-19-induced unemployment is associated with suicide and safety net utilization using administrative data.

Second, from a more general viewpoint, our study also contributes to the literature on the association between unemployment and suicide. This association has been extensively studied in social sciences [23–33] and public health (see a systematic review [34]) and many studies have shown that increases in unemployment are associated with a rise in the suicide rate. For example, Ruhm [23] shows that a one-percentage-point increase in the state unemployment rate was associated with an approximate 1.3% increase in the suicide rate in the US and some studies reviewed in Franquiho et al. [34] provide counterpart estimates with a range of 0.79–4.5% increases. These estimated percentages in the previous studies are much lower than our counterpart percentage of 37.4%, but one caveat is that our result is based on the largest monthly, not yearly as in most studies, estimate in the second and third quarters of 2020.

Third, our study is also related to the literature on the association of unemployment with poverty and safety net use. Some studies examine relations between unemployment-rate fluctuation and use of safety net programs such as unemployment insurance benefits and means-tested programs before the COVID-19 crisis [35–37]. Our findings add new evidence to the literature by estimating and comparing the association between unemployment and the use of multi-tier safety net programs.

The rest of the paper is organized as follows. Section 2 describes background information about the first wave of COVID-19 in Japan in 2020, provides the definitions and the descriptive statistics of our dataset, and presents some descriptive evidence regarding our research topic. Section 3 describes our research design and empirical model. In Section 4 we present baseline estimation results and in Section 5 we also provide the results of robustness checks and alternative estimations with a different employment-shock variable. Section 6 concludes.

## 2 Background and data

This section provides some background about the first wave of COVID-19 in Japan and the data that we use for our empirical analysis. First, we briefly describe the COVID-19 crisis in Japan, particularly focusing on the period of the first wave between April and June 2020. Second, we explain the definitions and institutional backgrounds of outcome and treatment variables. Third, we provide data sources and descriptive statistics of key variables. Fourth, we show descriptive evidence regarding the association between COVID-19-induced employment shocks and the outcomes of interest using time-series graphs, a bar plot, and scatter plots.

### 2.1 The first wave of COVID-19

During and after the first wave of COVID-19 in Japan, which started in March 2020 and ended in June 2020 (S1 Fig), Japan suffered from economic and social difficulties despite its

relative success in mitigating COVID-19 infection spread. Unlike many Western countries where COVID-19 lockdowns were legally enforced, in Japan, social distancing interventions including Japan's first "COVID-19 state of emergency" (hereafter the first state of emergency) were based on non-compulsory requests made by the central and local governments. However, several mobility indices show that mobility significantly decreased during the period of the first state of emergency (see, for example, S2 Fig). Subsequent reduction in social and economic activities under the first wave had considerable impact on employment.

At the same time, the period from the outset of the COVID-19 crisis in February 2020 to the end of the first wave of COVID-19 in June 2020 was an important stage in Japan's COVID-19 policy formation during which basic epidemiological, social and fiscal schemes against COVID-19 were constructed (S1 Fig). The first COVID-19 state of emergency was declared on 7th April in seven prefectures, including the Tokyo and Osaka metropolitan areas. On April 16, the State of Emergency Declaration was extended to cover the entire country. Then, as the infection level steadily dropped, the government gradually lifted the state of emergency in multiple phases starting in the middle of May.

During this period, the first and second supplementary budgets as emergent COVID-19 fiscal measures passed on 30th April and 12th June respectively. The total amount of these two supplementary budgets was 57,602 billion JPY (around 520 billion USD given that 1 USD = 110 JPY) and around 10.7% of GDP in 2020 [38]. The largest component of the first supplementary budget was a lump-sum transfer program consisting of a one-time transfer of 100,000 JPY (around 910 USD) to all individuals living in Japan. There were also several other transfer programs that were intended to support firms and their employees under the COVID-19 crisis, but financial supports for people who lost their jobs or incomes due to COVID-19 during this period were mostly realized by the existing three tiers of safety net programs that we explain in the next subsection. See Ando et al. [38] for more detailed background information and the contents of Japan's fiscal responses against the first wave of COVID-19 and the subsequent first COVID-19 state of emergency.

## 2.2 Outcomes of interest

Our outcomes of interest are the suicide rate by gender and the utilization of the three tiers of safety net programs before and after the first wave of COVID-19 in Japan. They consist of the monthly panel data of 47 prefectures from January 2018 to September 2020 (33 months)—47 prefectures, such as Tokyo, Kyoto, and Hokkaido, are second-tier local governments while the third-tier local governments consist of 1,718 municipalities. The full sample size is 1,551, although some safety-net outcome variables lack data in some months. We use the administrative aggregated statistics provided by the central government for all of the outcome variables. See also S1 Table for further details about the three tiers of safety net programs such as eligibility, amount per recipient, and duration.

**2.2.1 Suicide rate.** Our primary outcome of interest is the suicide rate, which is the number of suicides per 100,000 population. We investigate the monthly panel data of three suicide-rate variables: total suicide rate, female suicide rate, and male suicide rate. The number of suicides is aggregated based on the dates and places of suicide. Suicide statistics are originally provided as police statistics and aggregated and arranged by the Ministry of Health, Labour, and Welfare (MHLW) as the Statistics of Suicide (*Jisatsu no Tokei*).

**2.2.2 Unemployment benefits (first-tier safety net).** The first outcome of a safety net program examined is the number of unemployment benefit recipients per 100,000 population under the unemployment insurance program. As unemployment benefit recipients we count

the number of the people who received the "basic benefit" that is the main component of the Japanese unemployment insurance system.

Unemployment insurance is unarguably the first-tier safety net program for the unemployed in most developed countries, but there are some caveats about the Japanese unemployment insurance system. First, the coverage rate of the unemployment benefits among unemployed was less than 30% in Japan in the early 2010s and it is lower than other developed countries even after taking into account institutional differences [39]. In addition, the pseudo-coverage rate of unemployment benefits provided by OECD was 22.4% in Japan in 2018, which is also among the lowest in OECD countries. Note that the numerator of this rate is the number of beneficiaries of unemployment insurance and of non-contributory benefits for job seekers and the denominator is the number of unemployed individuals (over 15 years old) according to the ILO definition (see OECD website for Social Benefit Recipients (SOCR) annual data for further details). Second, the COVID-19-induced employment shocks are more concentrated on part-time or contingent workers [18] and contingent workers are often ineligible for unemployment benefits. As a result unemployment insurance may not have worked well as a social safety net against the COVID-19 employment shocks [40].

In addition, instead of original monthly outcomes, we use the year-on-year monthly outcomes in which an outcome value at month $t-12$ is subtracted from an outcome value at month $t$. This procedure of year-on-year difference is meant to capture a monthly change in the number of beneficiaries from one year earlier rather than a monthly total number, which is the sum of existing and new beneficiaries. This procedure also eliminates prefecture-specific monthly fixed effects.

**2.2.3 Temporary loan programs (second-tier safety net).**   The second and third outcomes of safety net programs are the numbers of accepted applications per 100,000 population for two types of interest-free and guarantor-free temporary loan programs: Emergency Small Amount Funds for those who urgently need cash (up to 200,000 JYP or about 1,800 USD) and General Support Funds for those who need cash for a certain period (up to 150,000–200,000 JYP or about 1,400–1,800 USD per month). They are both existing means-tested temporary loan programs which can be utilized as second-tier safety net programs before applying for public assistance. They were rarely used before the COVID-19 crisis as shown in Section 2.5. However, their requirements have also been relaxed since March 2020 and the number of loan recipients has increased dramatically.

**2.2.4 Housing Security Benefit (second-tier safety net).**   The fourth safety net program outcome is the number of newly accepted applications per 100,000 population for the Housing Security Benefit, which is another existing second-tier safety net program of short-term (i.e., three to nine months in 2020) housing allowance. This benefit scheme, originally intended only for those who have lost their jobs, was rarely used before the COVID crisis as shown in Section 2.5. But since April 2020, its eligibility has been extended to those who have not lost their jobs but have nevertheless experienced a large income reduction, resulting in a considerable increase in the number of households receiving these benefits. This housing allowance scheme covers the entire rent up to a certain upper limit.

**2.2.5 Public assistance (third-tier safety net).**   The fifth and sixth variables of safety net programs are the per-capita numbers of public assistance recipients and recipient households. The public assistance program in Japan, the third-tier safety net, is considered to be the "final safety net" or "last resort," and the prerequisites for application are in general strict, such as having no savings and no assets. We can use only total recipient and total recipient household numbers for monthly prefecture-level data.

As we use year-on-year monthly outcomes of unemployment benefit recipients, we also use year-on-year monthly public assistance outcomes. This procedure is meant to capture a

monthly change in the number of beneficiaries from one year earlier and to eliminate prefecture-specific monthly fixed effects.

## 2.3 Treatment variables

For a treatment variable, we use a continuous variable that reflects a sudden and plausibly exogenous regional variation in the COVID-19-induced employment shock. We adopt the unemployment rate as a baseline employment indicator because it is most commonly used in the literature. We also use a supplementary employment indicator based on the statistics of registered full-time-job seekers for a robustness check.

**2.3.1 Baseline employment shocks.** As a baseline treatment variable, we use a continuous variable that reflects a sudden and plausibly exogenous regional variation in the COVID-19-induced employment shock, using quarterly unemployment statistics. We define the COVID-19-induced employment shock as the following de-trended increase in the regional unemployment rate in the second quarter (i.e., April to June) of 2020:

$$EmpShock_i = [(X_{i,2020Q2} - X_{i,2019Q2}) - (X_{i,2019Q4} - X_{i,2018Q4})], \tag{1}$$

where $i$ indicates prefecture, $X$ is the unemployment rate, and $Q2$ and $Q4$ are the second and fourth quarters respectively. The first term in Eq (1) is a year-on-year difference in the unemployment rate in the second quarter of 2020, which consists of a COVID-19-induced employment shock and a prefecture-specific year trend. The second term is a corresponding year-on-year difference in the fourth quarter of 2019, which should reflect a prefecture-specific year trend just before the COVID-19 crisis. Assuming that the prefecture-specific year trends are similar just before and after the COVID-19 crisis, the difference between these two terms is thus expected to capture only the COVID-19-induced employment shock.

Note that what we use for $X_{i,t}$ is the prefecture-level estimates of the unemployment rate provided in the Labour Force Survey. The prefecture-level estimates are estimated by the Statistics Bureau. The Bureau states that some imprecision is expected in these estimates, but we treat the estimated values as true values and do not consider their statistical uncertainty in our statistical analysis due to data limitations.

**2.3.2 Alternative "full-time" employment shocks.** As an additional analysis, we construct another employment-shock treatment variable based on a more narrowly defined employment indicator. That is, for the numerator of the regional unemployment rate $X_{i,t}$ in Eq (1) we use the number of the unemployed people who are registered as full-time-job seekers at public employment security offices, instead of the total number of the unemployed. We call this variable the variable of a "full-time" employment shock and describe estimation results based on this alternative treatment variable in the section on robustness checks.

Because registered full-time-job seekers are often, though not always, the breadwinners of households and are eligible for unemployment benefits, this alternative "full-time" employment-shock variable may reflect a specific part of the entire employment shock in a prefecture. That is, this "full-time" employment-shock variable may strongly reflects the COVID-19 shocks for full-time workers who are entitled to receive unemployment benefits (i.e. to receive unemployment benefits, an unemployed person needs to be registered at public employment security offices). Note also that although $X_{i,t}$ is quarterly in the baseline definition of Eq (1), we use the number of the registered full-time-job seekers in June for the second quarter and December for the fourth quarter, because the year-on-year total numbers of registered job seekers temporarily dropped in April and May 2020, probably due to the COVID-19 state of emergency in these months.

**Table 1. Summary statistics.**

|  | N | Mean | Std.Dev. | Min | Max |
|---|---|---|---|---|---|
| **Treatment** | | | | | |
| Employment shock (baseline) | 47 | 0.25 | 0.42 | -0.77 | 1.23 |
| Employment shock (full-time) | 47 | 0.08 | 0.08 | -0.14 | 0.22 |
| **Outcome** | | | | | |
| Suicide rate (total) | 1551 | 1.36 | 0.38 | 0.18 | 3.27 |
| Suicide rate (female) | 1551 | 0.78 | 0.34 | 0.00 | 2.17 |
| Suicide rate (male) | 1551 | 1.97 | 0.63 | 0.00 | 4.69 |
| Unemployment benefit (total) | 1551 | 326.38 | 61.24 | 202.11 | 526.63 |
| Unemployment benefit (female) | 1551 | 375.74 | 73.20 | 225.50 | 607.71 |
| Unemployment benefit (male) | 1551 | 273.12 | 51.88 | 176.70 | 454.55 |
| Unemployment benefit (total, yoy) | 1551 | 12.25 | 38.01 | -137.25 | 191.96 |
| Unemployment benefit (female, yoy) | 1551 | 9.55 | 39.78 | -175.60 | 206.22 |
| Unemployment benefit (male, yoy) | 1551 | 15.20 | 38.51 | -94.49 | 187.69 |
| Emergency Small Ammount Funds | 893 | 28.51 | 55.85 | 0.00 | 567.45 |
| General Support Funds | 893 | 14.90 | 36.93 | 0.00 | 426.29 |
| Housing Security Benefit | 423 | 6.13 | 9.49 | 0.00 | 81.26 |
| Public assistance (recipients) | 1551 | 1398.27 | 666.37 | 332.39 | 3248.44 |
| Public assistance (households) | 1551 | 1103.91 | 514.65 | 289.87 | 2516.68 |
| Public assistance (recipients, yoy) | 1551 | -10.96 | 18.90 | -61.43 | 46.46 |
| Public assistance (households, yoy) | 1551 | 3.01 | 11.85 | -21.45 | 50.50 |
| **Covariate** | | | | | |
| Cumulative infection rate | 47 | 8.51 | 8.56 | 0.00 | 44.72 |
| Cumulative death rate | 47 | 0.43 | 0.63 | 0.00 | 2.37 |
| Google Mobility index | 47 | -22.18 | 4.86 | -39.05 | -12.87 |
| Population density | 47 | 1350.74 | 1781.23 | 234.70 | 9792.90 |
| Ratio of employees (secondary) | 47 | 0.25 | 0.05 | 0.14 | 0.33 |
| Ratio of employees (service) | 47 | 0.66 | 0.04 | 0.60 | 0.73 |
| Elderly dependency rate | 47 | 53.50 | 7.51 | 35.00 | 70.10 |
| Total population | 47 | 268.49 | 277.93 | 56.00 | 1392.00 |

Notes: The employment shock is a cross-section variable calculated based on Eq (1). All the outcome variables are calculated per 100,000 population. For the definition of each variable, see S2 Table. Outcome variables such as suicide rates, unemployment benefits, and Public assitance recipients are 21-months data (from January 2019 to September 2020) while the variables of Emergency Small Amount Funds, General Support Funds, and Housing Security Benefit are more restricted due to data limitation. "yoy" means year-on-year difference. Sources: See S2 Table.

## 2.4 Summary statistics

Table 1 shows the summary statistics of our dataset. Because the employment-shock variable defined in Eq (1) and covariates are time-invariant and cross-sectional variables, we have 47 observations. The outcome variables such as suicide rates, unemployment benefit recipients, and public assistance recipients include 33-months of data (from January 2018 to September 2020) while the variables of the second-tier safety net programs such as Emergency Small Amount Funds, General Support Funds, and Housing Security Benefit are more restricted due to data limitations.

For Emergency Small Amount Funds and General Support Funds, we use the monthly data between January 2019 and September 2020, but due to limited data availability we do not have the data for February and March 2020; statistics for these two months were missing from the

original data provided by the government. In addition, the monthly-level data aggregation from April to July 2020 is slightly irregular, although it should not cause serious estimation bias. More concretely, the number of loan decisions between April and September 2020 in the statistics provided by the central government is aggregated as follows: April data is based on the numbers from 25th March to 2nd May, May data is from 3rd May to 30th May, June data is from 31st May to 27th June, July data is from 28th June to 1st August, August data is from 2nd August to 29th August, and September data is from 30th August to 3rd October. For Housing Security Benefit, we can use only January, February, and March 2019 in the pre-COVID-19 period because of the limited data availability.

S2 Table presents definitions and data sources of all the variables used for analyses. All the data except for the second-tier safety net programs (Emergency Small Amount Funds, General Support Funds, and Housing Security Benefits) are obtained from the websites of the Ministry of Health, Labour, and Welfare (MHLW) and Statistics Bureau of Japan. When it comes to the data of the second-tier safety net programs, we use the datasets that are directly provided by MHLW.

## 2.5 Descriptive evidence

Before moving on to the econometric analysis, we provide some descriptive evidence that suggests an association between the increase in the unemployment rate and the increases in suicide and safety net use in the midst of the COVID-19 crisis. We firstly provide the nation-level time-series graphs of employment status and outcome variables before and after the onset of the first wave of COVID-19. We then present the scatter plots of suicide rates versus our baseline treatment variable (the COVID-19-induced employment shock measured by the unemployment rate).

Fig 1 describes how the status of employment, suicide, and safety-net provisions in Japan evolved before, during, and after the first wave of COVID-19 in 2020. These nation-level statistics are constructed based on four major employment statistics (labor force participation rate, employment rate, unemployment rate, and jobs-to-applicants ratio) and the outcome variables listed in Section 2.2. See also S2 Table for their definitions and data sources.

The implications of Fig 1 are summarized as follows. First, panels (a) and (b) show that the employment rate and labor force participation rate have sharply dropped since April 2020, while the unemployment rate and the jobs-to-applicants ratio—the ratio of job offers to job applicants, which is one of the most used monthly administrative employment indicators in Japan—gradually deteriorated.

Second, panels (c) and (d) suggest that suicide rates have been increasing since July 2020. It is of course not clear from these graphs to what degree the COVID-19 crisis has led to the increase in suicides, but some specialists and reports in the mass media have asserted that the sharp deterioration in employment in the first few months of the COVID-19 crisis may have resulted in the slightly lagged increase in suicides [41, 42]. In S3 Table, we also present the numbers of suicides in 2019 and 2020 by sex, age and occupation. These data show that the increase in suicides from 2019 and 2020 stands out in both employed and non-employed women within all age cohorts and young employed men aged 20–29.

Third, panels (e) and (f) show that the utilization of the first and second-tier safety net programs also increased during and after the second quarter of 2020 or the period of the first wave of COVID-19. Outcome values for all the second-tier programs are not zero in the pre-COVID-19 period but discontinuously smaller than those in the COVID-19 period. Note also that the central government has relaxed eligibility requirements for these second-tier programs due to COVID-19 since March 2020.

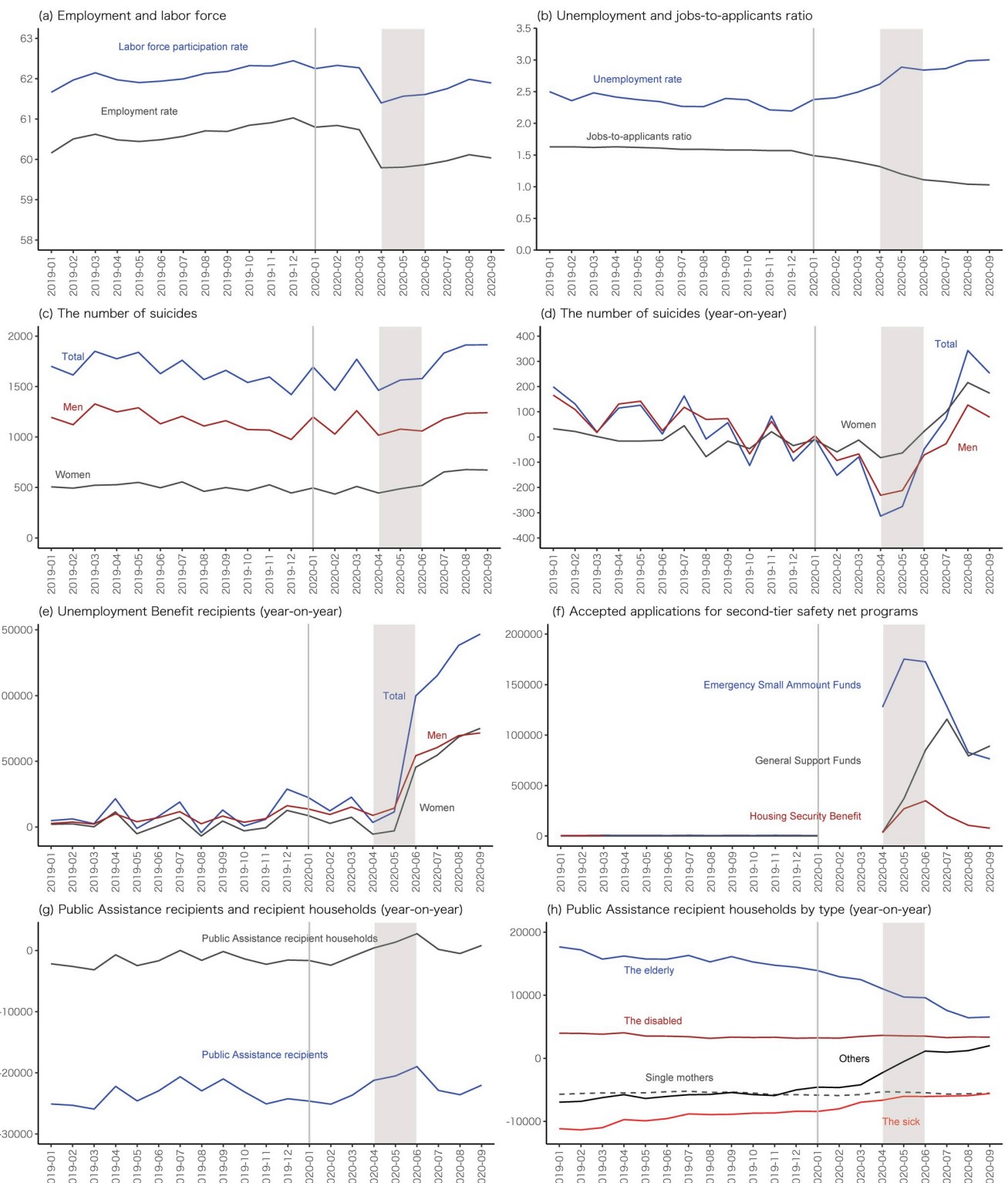

**Fig 1. Monthly trends of employment-related variables, suicides and safety-net programs in Japan.** Notes: A vertical line in January 2020 indicates the beginning of the COVID-19 crisis in Japan and a gray area indicates the second quarter of 2020. "Year-on-year" means year-on-year difference. Panels (a) and (b) show different employment-related statistics. Suicides in panels (c) and (d) are counted based on the estimated dates of death. In panel (f), "Emergency Small Amount Funds" and "General Support Funds" are two public temporary loan programs and "Housing Security Benefit" is a temporary housing allowance program. In panel (h), "the elderly" indicates a household with only persons aged 65 or over (and unmarried persons under the age of

18), "single mothers" means households with unmarried women and their children under the age of 18, "the disabled" or "the sick" are households the heads of which are persons with a disability or illness respectively. See Section 2 for more details about these data. Sources: See S2 Table.

Fourth, panel (g) shows that the year-on-year numbers of recipients and recipient households of the public assistance program, the third-tier and last-resort safety net, are negative both before and after the onset of the COVID-19 crisis, implying that the number of public assistance recipients kept decreasing regardless of the COVID-19 crisis. Panel (h), however, indicates that the counterpart year-on-year number for the household type of "others", which includes low-income workers and the unemployed, clearly started increasing in April 2020.

In turn, Fig 2 shows the regional variation of the employment shock measured as Eq (1) (panel (a)), namely our treatment variable, and correlations of this employment shock and changes in total, female, and male suicide rates in July 2020 from those in July 2019 (panel (b)-(d)). We use the monthly suicide rates in July because our employment-shock variable is calculated as a shock in the second quarter (i.e., April to June) 2020.

Panel (a) of Fig 2 shows that the most affected prefectures are Okinawa, Kanagawa, and Osaka and their employment shocks are more than one percentage point. Panel (a) also suggests that these employment shocks are larger in metropolitan areas (Kanagawa, Osaka, Nara, Tokyo, Hyogo, Saitama, Chiba, Kyoto) and prefectures with popular sightseeing spots (Okinawa and Hokkaido). Panels (b)-(d) present some positive correlations between the employment shock and the suicide rates, although suicide rates in the metropolitan areas are not necessarily highest.

## 3 Research design

Exploiting the regional variation in the COVID-19-induced employment shock and a difference-in-difference research design, we examine how the intensity of the regional employment shock is associated with the regional trends of suicide and the use of safety net programs in the COVID-19 period. In an estimation model, we also incorporate several observed covariates that may cause estimation bias if they are not controlled for.

### 3.1 Empirical model

The baseline model specification takes the following event-study specification with time-varying DID coefficients:

$$Y_{it} = \sum_{\tau \neq Jan.2020} \beta_{\tau} EmpShock_i \times 1[t = \tau] + \pi_i + \theta_t + \phi_i t + \varepsilon_{it}, \tag{2}$$

where $Y_{it}$ is the outcome variable such as a suicide rate for prefecture $i$ at time $t$, $Empshock_i$ is a continuous treatment variable of the COVID-19 employment shock defined as in Eq (1), $1[t = \tau]$ is a dummy variable that takes the value of one if $t = \tau$ and zero otherwise, $\pi_i$ and $\theta_t$ are prefecture and month fixed effects, respectively, $\phi_i t$ is an individual (i.e., prefecture) linear trend, and $\varepsilon_{it}$ is an error term.

The coefficients of interest are the time-varying coefficients $\beta_{\tau}$, which capture the correlation between $EmpShock_i$ and the outcome trend from January 2020 to time $\tau$. When time $\tau$ is before January 2020, $\beta_{\tau}$ can be interpreted as a placebo estimate that is expected to be around zero if no confounding trends existed before the COVID-19 crisis. When month $\tau$ is after January 2020, $\beta_{\tau}$ captures the association between $EmpShock_i$ and the outcome trend in the COVID-19 period, which is not confounded by pre-existing outcome trends if placebo estimates before January 2020 are around zero. Furthermore, if the assumption of no differential

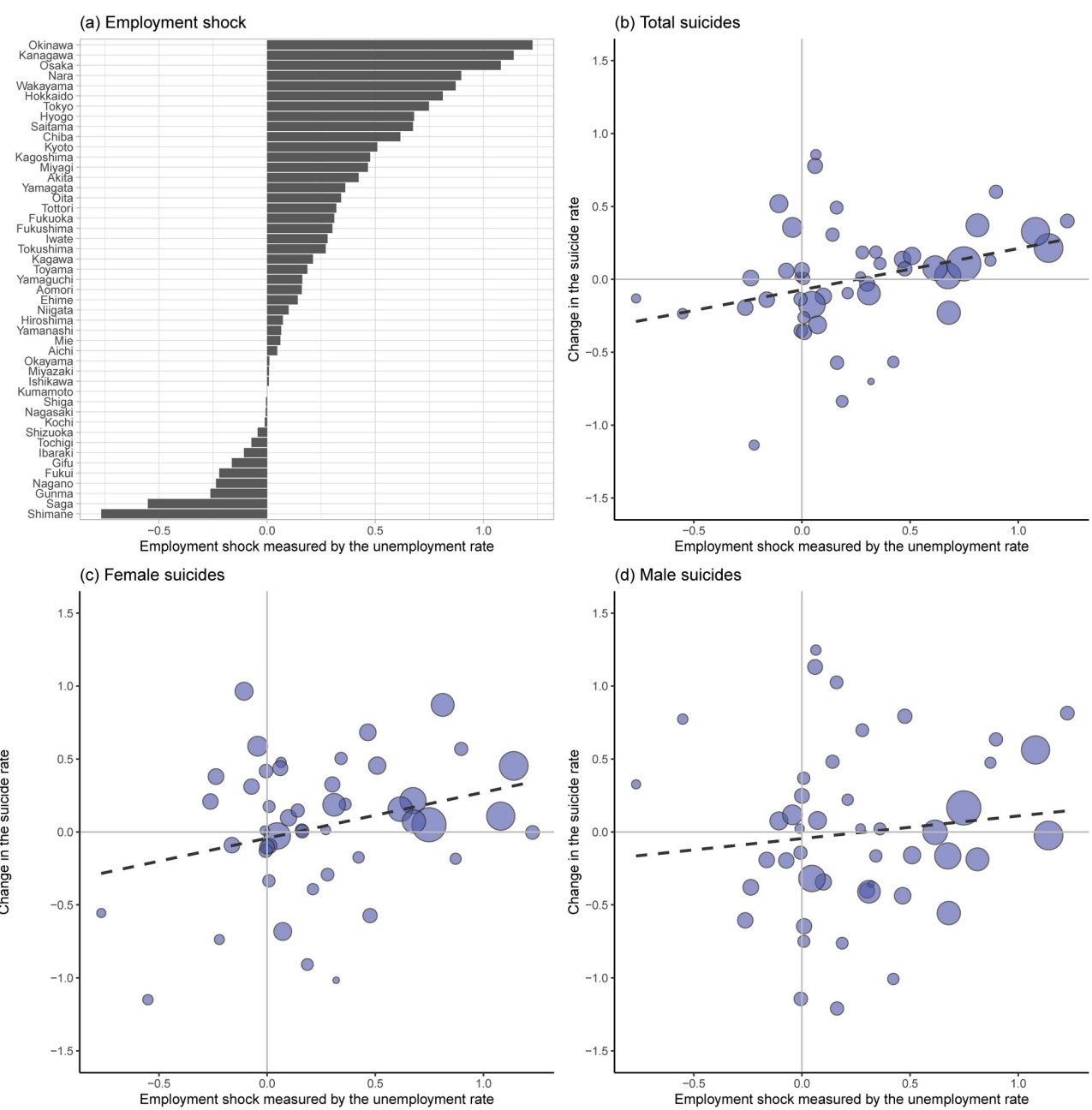

**Fig 2. Changes in the unemployment rate and suicide rates.** Notes: In all four graphs, the *X* axis is the size of an employment shock defined as in Eq (1). The *Y* axis in panels (b)-(d) is a change in the suicide rate from July 2019 to July 2020. The size of each circle is based on the population size of each prefecture. The dashed line is the fitted linear regression line based on the Ordinary Least Square (OLS) method. Sources: See S2 Table.

trends among prefectures in the COVID-19 period is plausible, $\beta_\tau$ in the COVID-19 period can be interpreted as a causal effect of $EmpShock_i$ on $Y_{i\tau}$. We however do not assert such a clear causal interpretation due to possible unobserved confounders in the COVID-19 period.

In the baseline regression analysis we use the weighted least square (WLS) estimation method in which prefecture-level population sizes are used as weights. With this WLS model, the heterogeneous association of the employment shock with an outcome for a larger

prefecture is more highly weighted in a DID estimator. In the section of robustness checks, we also use the ordinary least square (OLS) estimation method because it is not clear whether a WLS estimator is more appropriate than an OLS estimator as an estimator of a population average effect when effect heterogeneity exists [43].

In addition, for the outcomes of the second-tier safety net programs (two temporary loan programs and Housing Security Benefit), we do not include an individual linear trend $\phi_i t$ in Eq (2) because the levels of these outcomes are negligible in the pre-COVID-crisis period compared with the COVID-crisis period (see panel (f) in Fig 1). As robustness checks, we also provide estimation results of all of the other outcomes with an estimation model that does not control for individual linear trends.

One caveat other than the problem of possible confounders, which we will discuss in the next subsection, is that what we examine is the region-level (i.e., prefecture-level) association between the employment shock and the outcomes of interest. In this sense, we do not directly investigate how individual or household-level unemployment is related to individual or household-level suicide and safety net use. Instead, we examine how the prefecture-level employment shock is associated with suicide and safety net use for people and households living in each prefecture. In other words, the parameter $\beta_\tau$ reflects not only direct associations between individual unemployment and individual outcomes, but broader associations between deteriorating prefecture-level employment situations and the well-being and living standard of people and families living in a prefecture.

### 3.2 Possible confounders

Even if placebo estimates of $\beta_\tau$ before January 2020 in Eq (2) are around zero, it is possible that some regional factors are correlated with both $EmpShock_i$ and $Y_{it}$ under the COVID-19 crisis, resulting in confounding bias in the estimation of $\beta_\tau$ after January 2020. Such confounding regional factors may exist given the fact that the impact of the COVID-19 crisis has spread through society via various pathways.

In order to mitigate such possible confounding bias, we also estimate the regression model that incorporates pre-determined cross-sectional covariates interacted with monthly dummy variables in the COVID-19 period. These terms capture the COVID-19-induced outcome changes that are better explained by pre-determined factors than by the COVID-19 employment shock. Time-varying covariates, which are more commonly used in the DID literature, cannot be obtained in our analysis with monthly panel data.

For the pre-determined covariates, we use the three variables that reflect the intensity of the COVID-19 crisis and the five variables that represent pre-COVID-19 demographic conditions. The three covariates of the COVID-19 crisis consist of the cumulative COVID-19 infection and death rates at the end of the first wave (i.e., at the end of June 2020 based on S1 Fig) and the monthly average of the Google Mobility index in May 2020, when the first COVID-19 state of emergency was imposed. All of these variables may affect both the employment shock and the outcomes of interest and may cause omitted variables bias. The five demographic variables are the population density measured by inhabitable area, the ratios of employees in the secondary and tertiary (i.e., service) industry, the population size, and the elderly dependency rate (i.e., the ratio between the number of persons aged 65 and over and the number of persons aged between 15 and 64). See Table 1 for the summary statistics of the eight covariates.

Regarding the Google mobility data, it is taken from Google COVID-19 Community Mobility Reports [44]. Google provides prefectural data on people's visits to six categories of places such as "Grocery and pharmacy", "Retail and recreation", "Parks", "Transit stations", "Workplace", and "Residential". Each indicator shows the percentage change in the number of

visitors to (or time spent in) different locations compared to the baseline number computed from January 3rd and February 6th, 2020. Using four of these mobility measures —"Grocery and pharmacy', "Retail and recreation", "Transit stations", and "Workplace" —, we calculated the monthly average values of Google Mobility indicators, following the definition of [45, 46]. See also S2 Fig for nation-level time-series statistics of this mobility index in 2020.

We do not strongly argue that our empirical strategy and the inclusion of the above eight covariates perfectly solve the problem of estimation bias. In this sense, we interpret our estimation results as associations between the COVID-19-induced employment shock and the outcomes of interest, which may reflect causal relations from the former to the latter.

### 3.3 Suicide prevention by safety net

Another limitation of our research design is that we interpret safety net use as an outcome of financial distress rather than an indicator of financial aid. That is, although it is important to understand how the safety net programs have prevented suicide under the COVID-19 crisis, we do not address this question in this paper. This may be disappointing, but the incidences of suicide and safety net use are expected to be strongly endogenous and we cannot find any proper exogenous variation that can solve this endogeneity problem.

This limitation, however, does not lead to any confounding bias in our estimation of $\beta_\tau$ because the endogenous relationship between outcome variables is not relevant for our empirical strategy. We only need to interpret the association between the COVID-19-induced employment shock and the suicide rate as the one that remained even after the three tiers of safety net programs and other financial benefits contributed to suicide prevention during and after the first wave of COVID-19.

## 4 Results

This section provides baseline estimation results based on WLS regression with and without covariates. We present results for suicide rates and the three tiers of safety net programs separately.

### 4.1 Suicide rates

Fig 3 shows the estimation results for the suicide rates: the left-hand graphs are estimation results based on Eq (2) using WLS and the right-hand graphs show counterpart results based on a model that additionally controls for the eight covariates. First, placebo estimates in the pre-COVID-19 period are more or less zero, implying that there is no clear statistically significant correlation between the employment shock and all the outcome trends before January 2020. Second, after January 2020, DID estimates for total, female, and male suicide rates are positive from May to July and the lower limits of the 90% confidence intervals are often above or near zero, particularly in July.

Comparing results in the COVID-19 period between the models without the covariates (right-hand) and with the covariates (left-hand), estimation results are robust in the sense that the positive estimates in July 2020 remain the same. Given that the right-hand results are less biased in the sense that some possible confounding factors are controlled for, in the rest of this paper we discuss the results based on the right-hand graphs. We also interpret that larger confidence intervals in some estimates after controlling for the covariates are partly due to correlations among the treatment variable and the eight covariates: when we regress the treatment variable on the eight covariates using cross-sectional data, the adjusted R squared is 0.352. We nonetheless use all eight covariates because the Variable Inflation Factor (VIF) of the treatment variable is 1.87 and relatively low.

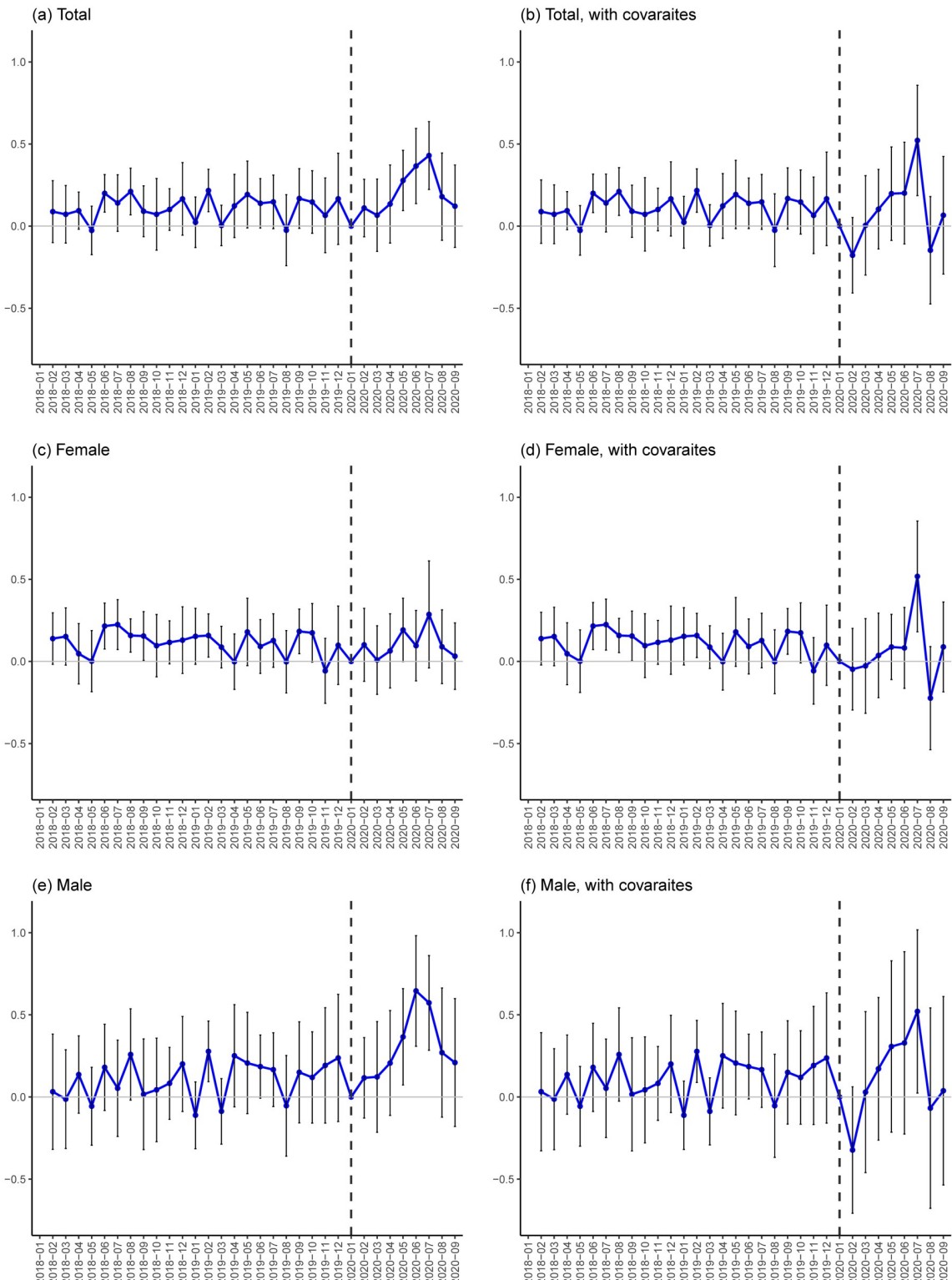

**Fig 3. DID estimates for suicide rates.** Notes: Each plot indicates a point estimate and a vertical line indicates a 90% confidence interval that is calculated based on a robust standard error clustered at the prefecture level. All the outcomes are measured as the number of suicides per 100,000 people and the treatment variable is the COVID-19-induced employment shock, which is calculated as Eq (1). Estimates are obtained based on Eq (2) with WLS estimation weighted by prefecture population size. Because Eq (2) incorporates individual (i.e. prefecture) linear trends, estimates can be obtained from February 2019. See S4 and S5 Tables for the values of these baseline estimates and standard errors in the COVID-19 period.

The sizes of the estimates in July 2020 based on the right-hand graphs suggest that the one-percentage-point increase in the COVID-19 employment shock in the second quarter of 2020 is associated with 0.522, 0.518, and 0.521-point increases (or 37.4%, 60.5%, and 26.5% increases from July 2019) in the total, female, and male suicide rates respectively. Note that the rates of increase are calculated as the estimates divided by the total, female, and male suicide rates in July 2019, which are 1.394, 0.856, and 1.963 respectively. In August and September, estimates are around zero, implying that suicides in these months may not directly be related to the second-quarter employment shock in 2020.

Overall, the DID estimates suggest that the COVID-19-induced employment shocks are clearly associated with both female and male suicides. Estimates are more or less robust to the inclusion of the covariates at least in July 2020, just after the second-quarter employment shock. This indicates that confounding bias may not be serious. At the same time, the association between the employment shock and the suicide rate is not clearly observed in August and September 2020 and this finding is also robust to the introduction of the covariates. This implies that the COVID-19 employment shock in the second quarter of 2020 may have a short-run association with suicide in the following month of July 2020, but cannot explain suicide increases in August and September 2020.

The magnitudes of the estimated associations in July 2020 are not small. Simple calculation based on the estimate for total suicides in panel (b) suggests that if a region with a population of 10 million experienced a one-percentage point increase in the unemployment rate caused by the COVID-19 crisis in the second quarter of 2020, which is roughly equivalent to the third-highest employment shock on Osaka (see Fig 2), this could have been associated with an additional 52.2 suicides in July 2020. Another simplified back-of-envelop calculation suggests that if Japan experienced the same one-percentage-point employment shock and there was a homogeneous nationwide association between the employment shock and the estimated 0.522 point increase in suicides, this could have led to an additional 657 suicides when the counterpart monthly suicide number was 1,761 in July 2019. Note that 657 is calculated as 1,258 times 0.522, where 1,258 hundred thousand (i.e. 125.8 million) is Japan's estimated population in July 2020).

## 4.2 Safety net use

When it comes to the relation between the COVID-19-induced employment shock and safety net programs, we provide three figures that represent estimation results for the first, second, and third-tier safety net programs respectively.

First, Fig 4 provides estimation results for year-on-year unemployment benefit numbers, the first-tier safety net. As in Fig 3, the left-hand results are based on Eq (2) using WLS and the right-hand results are based on the model that controls for the eight covariates. Note that we also examine the original, not year-on-year, outcome but some differential pre-trends for this outcome are not properly controlled for.

To begin with, estimates are slightly increasing but not significantly different from zero in the COVID-19 crisis period if no covariates are introduced in the regression model (left graphs). However, after controlling for the covariates, estimates in the COVID-19 period get larger for all three outcomes, particularly for the outcome of female recipients (right graphs). In July 2020, the one-percentage-point increase in the employment shock is associated with approximately an additional 27.9 total unemployment benefit recipients per 100,000 population (panel (b)).

Second, Fig 5 provides estimates for the second-tier safety net programs: the two temporary loan programs and the Housing Security Benefit. In the COVID-19 period, panels (a) to (d)

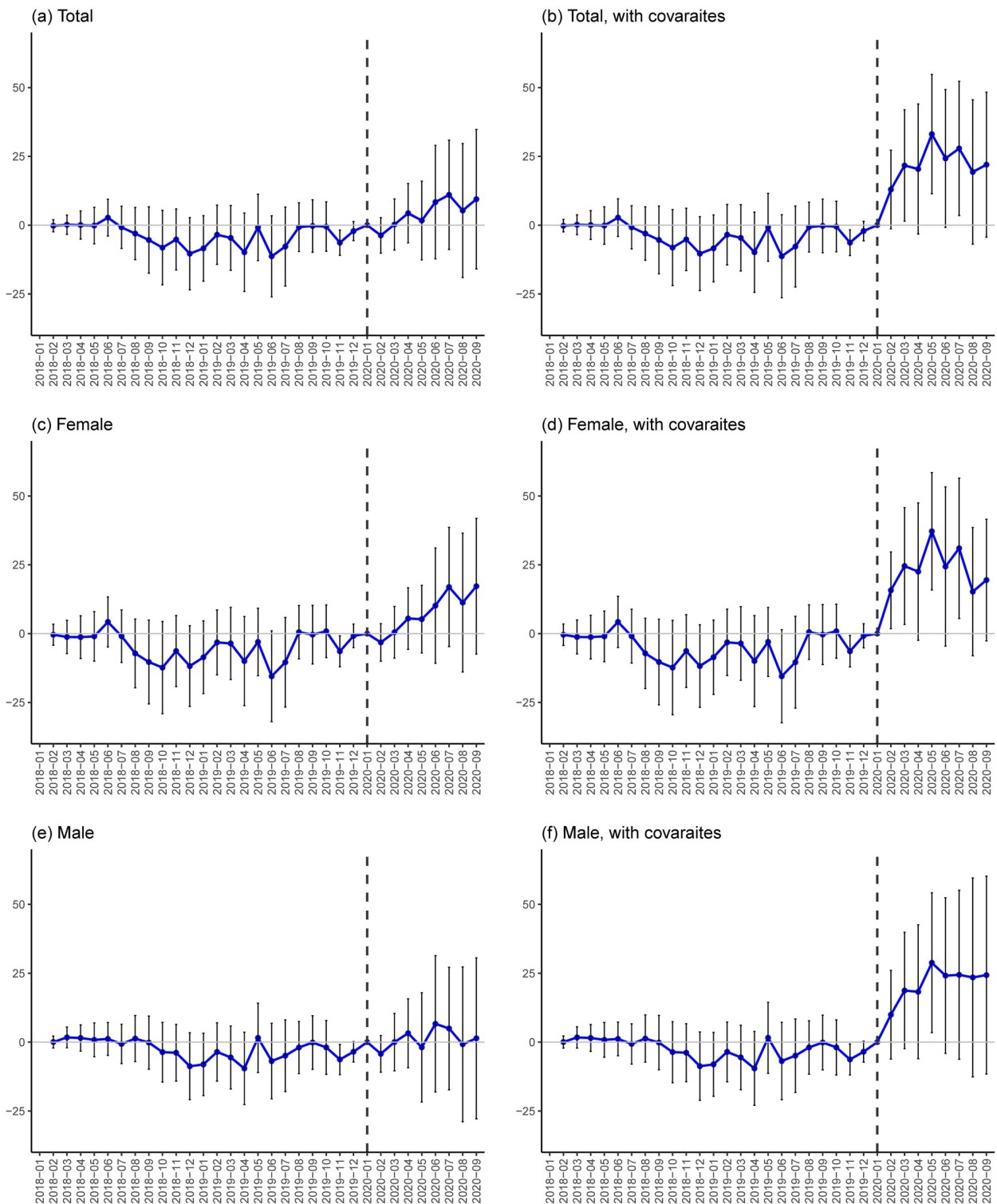

**Fig 4. DID estimates for unemployment benefit recipients.** Notes: Each plot indicates a point estimate and a vertical line indicates a 90% confidence interval that is calculated based on a robust standard error clustered at the prefecture level. All the outcomes are measured as the number of recipients per 100,000 people and the treatment variable is the COVID-19-induced employment shock, which is calculated as Eq (1). An outcome variable is year-on-year and calculated as the difference between an outcome value at month *t* and an outcome value at month *t*–12. Estimates are obtained based on Eq (2) with WLS estimation weighted by prefecture population size. Because Eq (2) incorporates individual (i.e. prefecture) linear trends, estimates can be obtained from February 2019. See S6 and S7 Tables for the values of these baseline estimates and standard errors in the COVID-19 period.

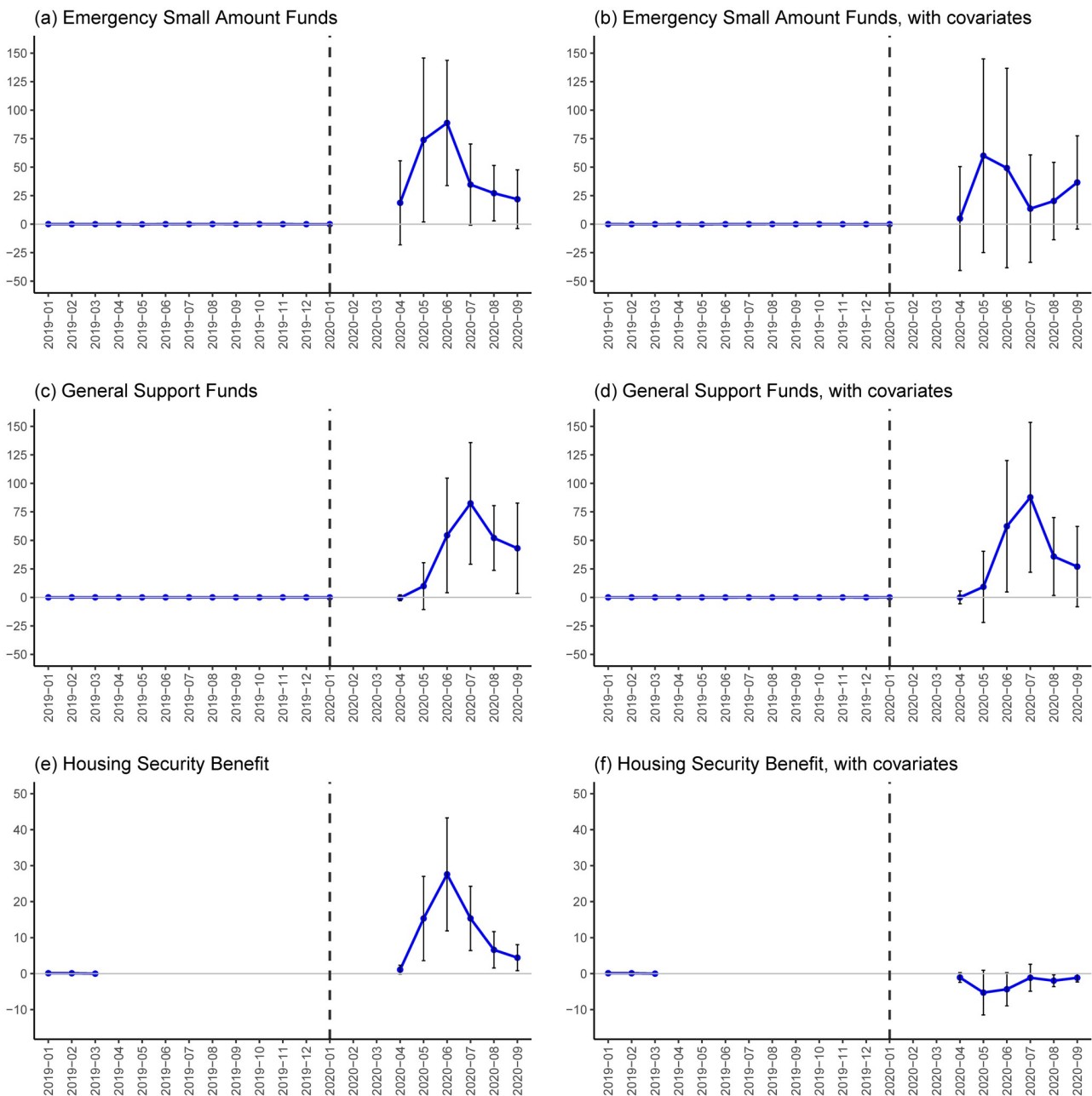

**Fig 5. DID estimates for second-tier safety net programs.** Notes: Each plot indicates a point estimate and a vertical line indicates a 90% confidence interval that is calculated based on a robust standard error clustered at the prefecture level. All of the outcomes are measured as the number of accepted applications per 100,000 people and the treatment variable is the COVID-19-induced employment shock, which is calculated as Eq (1). Estimates are obtained based on Eq (2) with WLS estimation weighted by prefecture population size. Individual linear trends are not introduced due to discontinuously smaller outcome values in the pre-COVID-19 period. For the outcome of Emergency Small Amount Funds and General Support Funds, data for February and March 2020 is missing. For the outcome of the Housing Security Benefit, the reference year is March 2019 instead of January 2020 due to the lack of data during April 2019 and March 2020. See S8 and S9 Tables for the values of these baseline estimates and standard errors in the COVID-19 period.

show that the estimates for the temporary loan programs in the COVID-19 periods are positive and these results are robust to the inclusion of the covariates, while the lower limits of the confidence intervals in panel (b) are below zero in the COVID-19 period. Panel (e) shows a positive association between the employment shock and Housing Security Benefit in the

COVID-19 period, but it disappears once the covariates are controlled for in panel (f). At the peak, the one-percentage-point increase in the employment shock is associated with an additional 87.8 accepted applications for General Support Funds in July 2020 (both per 100,000 population, panel (d)).

Note that the numbers of accepted applications for these programs are discontinuously smaller in the pre-COVID-19 period, so the pre-COVID-19 estimates and confidence intervals can be obtained but are negligibly smaller than those in the COVID-19 period. This may be one drawback of the examination of pre-trends for these outcomes, but it also means that there are at least no significant differential trends for these outcomes in the pre-COVID-19 period. We also do not incorporate individual linear trends due to this feature. As for the Housing Security Benefit, for which we have very limited prefecture-level monthly data in the pre-COVID-19 period (i.e., only January to March 2019), we at least know that the number of accepted applications discontinuously increased in 2020 at the national level: the average monthly number of accepted applications in fiscal year 2019 (from April 2019 to March 2020) in the whole of Japan was only 331, whereas the corresponding numbers in April, May, and June 2020 were 3,409, 27040, and 34,867 respectively (based on the statistics of MHLM).

Third, Fig 6 provides estimation results for year-on-year public assistance benefits: panels (a) and (b) present estimates for the number of recipients and panels (c) and (d) for the number of recipient households. All of the graphs show that estimates in the COVID-19 period are clearly increasing and positive while estimates in the pre-COVID-19 period are around zero. Introduction of the covariates leads to smaller estimates and larger confidence intervals in the COVID-19 period, but baseline findings remain the same—as already discussed, these larger confidence intervals may be explained by correlations among the treatment and the eight covariates. The one-percentage-point increase in the employment shock is associated with approximately an additional 9.7 and 11.6 public assistance recipients per 100,000 population in July and September 2020, respectively (panel (b)). Note that we also examine the original, not year-on-year, outcome but differential pre-trends are not properly eliminated for this outcome, particularly when individual linear trends are not incorporated.

Overall, these estimation results imply that the employment shock under the COVID-19 is positively associated with increases in safety-net utilization in all three tiers of programs in the COVID-19 period. The maximum estimate for each outcome suggests that a higher employment shock is associated with higher utilization. This association is stronger in the case of the two temporary loan programs in the second-tier of the safety net then in the case of the unemployment benefit, which is the first-tier program. Smaller estimates for unemployment-benefit utilization than those for the temporary loan programs may be explained by the fact that the unemployment-benefit coverage rate is low in Japan and that COVID-19-induced employment shocks are more concentrated on contingent workers who are often not eligible for unemployment benefits (Section 2.2).

In addition, even smaller estimates for the third-tier program of public assistance suggests that people facing unemployment and income reduction under the COVID-19 crisis tend to rely on the temporary loan programs rather than public assistance. Given the fact that these temporary loan programs were rarely used in the pre-COVID-19 period, the role of these programs as a safety net in the COVID-19 crisis stands out.

Finally, while the estimates for the temporary loan programs peaked in June or July 2020 (Fig 5), the estimates for public assistance continued to increase through the third quarter of 2020 (Fig 6). This may imply that a certain portion of the unemployed people who used the second-tier safety net programs had to move to public assistance, because the second-tier programs are temporary and their loan and benefit levels are rather limited.

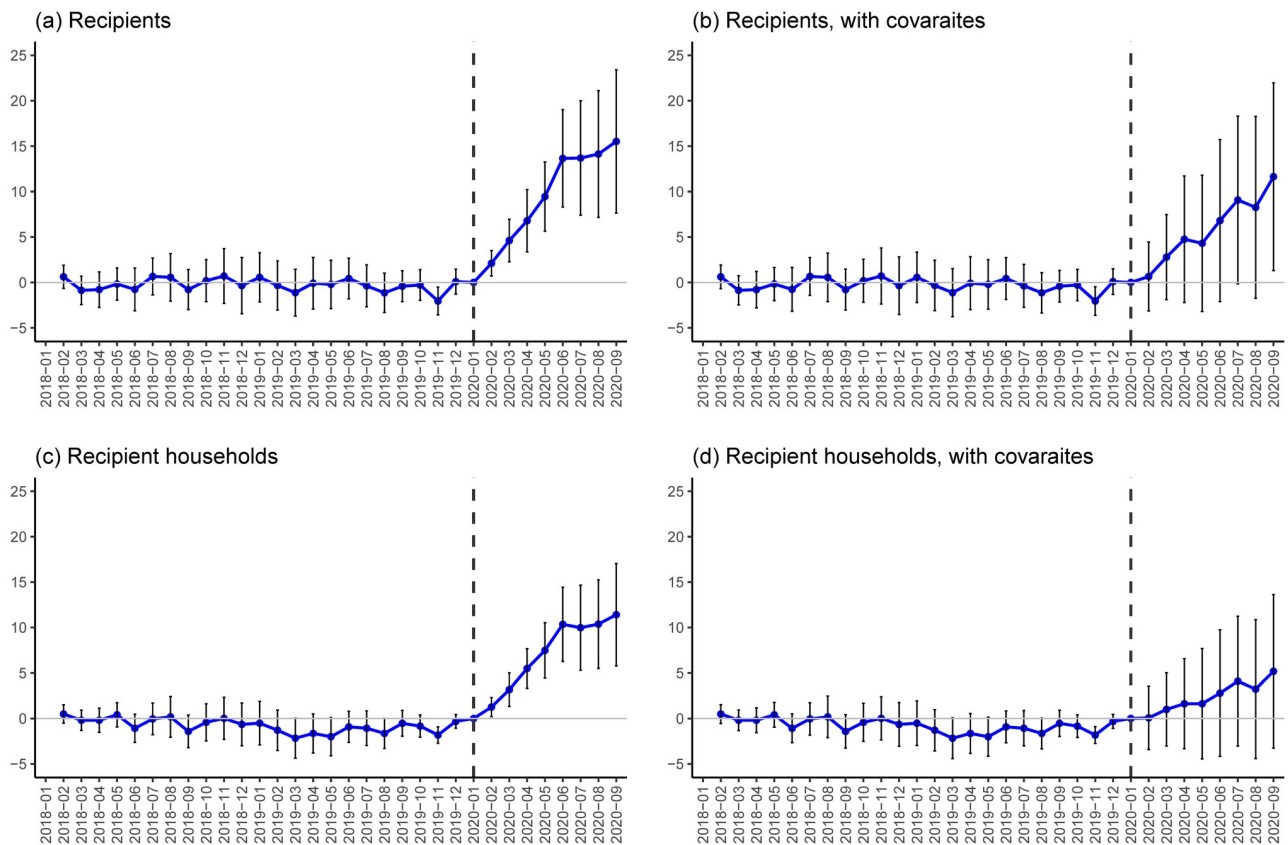

**Fig 6. DID estimates for public assistance.** Notes: Each plot indicates a point estimate and a vertical line indicates a 90% confidence interval that is calculated based on a robust standard error clustered at the prefecture level. All of the outcomes are measured as the number of recipients or recipient households per 100,000 people and the treatment variable is the COVID-19-induced employment shock, which is calculated as Eq (1). An outcome variable is year-on-year and calculated as the difference between an outcome value at month $t$ and an outcome value at month $t$–12. Estimates are obtained based on Eq (2) with WLS estimation weighted by prefecture population size. Because Eq (2) incorporates individual (i.e. prefecture) linear trends, estimates can be obtained from February 2019. See S10 and S11 Tables for the values of these baseline estimates and standard errors in the COVID-19 period.

## 5 Robustness checks

This section provides three different robustness checks. First, we implement regressions using different estimation settings while using the same treatment and outcome variables. Second, we change the reference period from January 2020 to all pre-COVID-19 months in the sample, namely January 2020 and earlier. Third, we re-analyze the same baseline models using an alternative treatment variable discussed in Section 2.3.

### 5.1 Weighting and individual trends

Our first robustness check is to present estimation results based on two regression weighting schemes (OLS or WLS) and two model specifications (with or without individual linear trends $\phi_i t$ in Eq (2)), using the same treatment and outcome variables as in the baseline estimation. The rationales for these robustness checks are as follows. First, as Solon et al. [43] argue, it is not clear a priori whether OLS or WLS regression is more suitable for estimating a population average effect, so we compare OLS and WLS estimates and discuss how they differ and why. Second, while incorporating individual linear trends may be effective for controlling for

observed and unobserved differential trends across prefectures, it is useful to examine whether estimation results change if we use a simpler model without individual linear trends.

Estimation results suggest that our primary findings for suicide and all the safety net programs are robust to these different estimation settings except for a few cases. First, S3 Fig shows that estimates for total, female, and male suicide rates in July 2020 are positive and mostly significantly different from zero regardless of estimation settings and the addition of the covariates. Second, S4 Fig illustrates that estimates for unemployment benefit recipients are also increasing during the COVID-19 period although some estimates for male unemployment benefit recipients are not significantly different from zero. Third, in S5 Fig, estimation results for the second-tier programs (i.e., temporary loans and Housing Security Benefits) do not change much under different weighting schemes; as in the baseline analysis we do not incorporate linear individual trends due to discontinuously smaller pre-COVID-crisis outcome values. Fourth, S6 Fig shows, once individual trends are controlled for, both OLS and WLS estimates for public assistance are increasing and positive in the COVID-19 period regardless of the introduction of the covariates, although standard errors are larger if the covariates are incorporated. If individual linear trends are not incorporated, pre-COVID-19 estimates for public assistance outcomes fail to be around zero and positive, implying that DID estimates in the COVID-19 period under this specification have a downward bias.

For most outcomes, WLS estimates during the COVID-19 period tend to be higher than the counterpart OLS estimates. We interpret larger WLS estimates as reflecting heterogeneous employment-shock associations with suicide across prefectures: as Fig 2 indicates, prefectures with larger populations may be more clearly associated with the employment shocks and the WLS estimation puts more weights on these prefectures.

## 5.2 A different reference period

We also provide a different robustness check in which all months before January 2020 are included in the reference period in addition to January 2020 as follows:

$$Y_{it} = \sum_{\tau > Jan.2020} \beta_\tau EmpShock_i \times 1[t = \tau] + \pi_i + \theta_t + \phi_i t + \varepsilon_{it}, \tag{3}$$

where the notation of $\tau \neq Jan.2020$ in Eq (2) is changed to $\tau > Jan.2020$. Because the pre-COVID-19 outcome trends are not correlated with the treatment variable in the baseline estimations based on Eq (2), including all pre-COVID-19 months in the reference period may improve precision by averaging out noise in monthly pre-COVID-19 outcomes without increasing estimation bias.

Supporting information provides the estimation results of this robustness check for the suicide rates (S4 and S5 Tables), unemployment benefit recipients (S6 and S7 Tables), second-tier safety net programs (S8 and S9 Tables), and public assistance (S10 and S11 Tables). In each table, we also provide the baseline estimates in the COVID-19 period that are presented in Figs 3–6 for comparison.

The estimation results of these robustness checks show that we robustly observe similar findings to those in the baseline estimations when we change the reference period from one pre-COVID-19 month to all pre-COVID-19 months. This implies that our results are not driven by specific pre-COVID-19 outcome values in January 2020. In particular, for the second-tier safety net programs, the inclusion of all pre-COVID-19 months in the reference period results in only subtle changes in estimates because their pre-COVID-19 benefit levels are discontinuously smaller than those in the COVID-19 period.

## 5.3 An alternative treatment

As a final robustness check, we present DID estimation results using an alternative treatment variable of the "full-time" employment shock described in Section 2.3. We provide estimation results based on all four estimation schemes presented in Section 5.1: OLS without linear trends, OLS with linear trends, WLS without linear trends, and WLS with linear trends.

Our estimation results can be summarized as follows. First, contrary to our main findings, estimates for suicide in the COVID-19 period are not significantly different from zero regardless of the addition of the covariates (S7 Fig). Second, estimation results for all three tiers of safety net programs are similar to the baseline results in the sense that estimates are often positive and significantly different from zero in the COVID-19 period (S8–S10 Figs). Significant decreases in estimates for Housing Security Benefit with the introduction of the covariates are also observed in this alternative analysis (S9 Fig). At the same time, estimates for unemployment benefit recipients are smaller when the covariates are incorporated (S8 Fig) and modest differential pre-trends are observed in the results for public assistance outcomes (S10 Fig). These results also suggest that the "full-time" employment shocks have different characteristics than the baseline employment shocks.

No significant association between the "full-time" employment shock and suicide may be explained by the fact that the baseline employment shocks and the "full-time" employment shocks capture partly different COVID-19-induced economic shocks. In fact, S11 Fig shows that the baseline employment shocks (X axis) are largest in prefectures in metropolitan areas or with popular sightseeing spots (e.g. Okinawa, Kanagawa, Osaka, Nara, Hokkaido, Hyogo, Tokyo, Saitama, Chiba, and Kyoto) whereas the alternative "full-time" employment shocks (Y axis) tend to be largest in prefectures with major manufacturing regions (e.g. Gifu, Aichi, Shizuoka, Shiga, Hiroshima, Toyama, and Mie). We speculate that there being no association between the "full-time" employment shocks and suicide is related to the fact that registered full-time-job seekers, who are taken into account in this alternative treatment variable, are more resilient to the risk of suicide. This may also be related to the fact that registered full-time-job seekers are often eligible for unemployment benefits, but further investigation is required.

## 6 Discussion and conclusion

Exploiting regional variations in the employment shocks caused by the COVID-19 crisis, this paper examines whether the COVID-19-induced employment shocks in the second quarter of 2020 are associated with increases in suicide and safety net use in the second and third quarters of 2020.

Our estimation results suggest that the COVID-19-induced increase in unemployment is associated with both suicide rates and safety net utilization. The sizes of the estimates are not socio-economically negligible. For example, let us consider a stylised region or prefecture with a population of 10 million where the COVID-19-induced economic shock is one percentage point in the second quarter of 2020 under the first COVID-19 state of emergency. In this region, this employment shock would then be associated with an additional 52 suicides, 2,790 recipients of the unemployment benefit, 8,780 recipients of a temporary loan program (i.e. General Support Funds), and 970 recipients of public assistance in July 2020.

In particular, the sizes of the estimates for suicide rates in our analysis are much higher than the corresponding previous estimates. Although previous studies use different data (mostly cross-country or cross-region yearly panel data) and different research designs (including correlational studies without explicit empirical strategies), our baseline estimation result of a 37.4% increase in the suicide rate associated with a one-percentage-point increase in

the unemployment rate is much higher than has been observed in the past. For example, the baseline estimate in Ruhm [23] is a 1.3% increase in the suicide rate. A systematic review of Fransquilho et al. [34] examines and summarizes many related studies, some of which show that a one-percentage-point increase in the unemployment rate is associated with a 0.79–4.5% increase in the suicide rate, although this review does not systematically compare estimation results in different studies.

There may be several reasons behind our larger estimates although direct comparisons with other studies are difficult. First, our study is primarily correlational rather than causal and there is a possibility that confounding upward bias causes such large estimates. The addition of the eight covariates does not change the estimates much, but it is still possible that such confounding bias exists. Second, the above comparisons with the previous studies are based on our maximum *monthly* estimate in July 2020, which can be larger than the *yearly* estimates that most previous studies present. To obtain a counterpart yearly averaged estimate is impossible or misleading because of the short-term feature of our employment-shock treatment variable. Third, one simple interpretation is that the sharp employment shock in the second quarter of 2020 under the COVID-19 crisis may have resulted in unusual short-term impact on suicide and suicidal ideation, likely in interaction with other factors. This interpretation may be plausible, but more rigorous analysis is required to determine the plausibility of this interpretation.

In conclusion, our findings of association between the employment shock during the first wave of COVID-19 and suicide and safety net use implies that the increase in COVID-19-related unemployment has been associated with non-negligible increases in suicide and financial distress. At the same time, due to the limitations of our aggregated data and the scope of our research design, this paper cannot disentangle more precisely the interplay between unemployment, mental and financial distress, and safety net participation under the COVID-19 crisis and its causal mechanisms. These topics should be examined in future studies.

## Supporting information

**S1 Fig. Confirmed cases/deaths and government responses in Jan.-Jun. 2020.**
(PDF)

**S2 Fig. Mobility during and after the first COVID-19 state of emergency.**
(PDF)

**S3 Fig. Additional DID estimates for suicides.**
(PDF)

**S4 Fig. Additional DID estimates for unemployment benefit recipients.**
(PDF)

**S5 Fig. Additional DID estimates for second-tier safety net.**
(PDF)

**S6 Fig. Additional DID estimates for public assistance.**
(PDF)

**S7 Fig. DID estimates for suicides ("full-time" employment shock).**
(PDF)

**S8 Fig. DID estimates for unemployment benefit recipients ("full-time" employment shock).**
(PDF)

**S9 Fig. DID estimates for second-tier safety net ("full-time" employment shock).**
(PDF)

**S10 Fig. DID estimates for public assistance ("full-time" employment shock).**
(PDF)

**S11 Fig. Correlation between employment-shock variables.**
(PDF)

**S1 Table. Description of the three tiers of safety net programs.**
(PDF)

**S2 Table. Variable definitions and data sources.**
(PDF)

**S3 Table. Suicides in 2019 and 2020 by age and occupation.**
(PDF)

**S4 Table. Estimation results for suicide rates, without covariates.**
(PDF)

**S5 Table. Estimation results for suicide rates, with covariates.**
(PDF)

**S6 Table. Estimation results for unemployment benefits, without covariates.**
(PDF)

**S7 Table. Estimation results for unemployment benefits, with covariates.**
(PDF)

**S8 Table. Estimation results for second-tier safety net, without covariates.**
(PDF)

**S9 Table. Estimation results for second-tier safety net, with covariates.**
(PDF)

**S10 Table. Estimation results for public assistance, without covariates.**
(PDF)

**S11 Table. Estimation results for public assistance, with covariates.**
(PDF)

**S1 Appendix. All the supporting information in one PDF file.**
(PDF)

## Acknowledgments

This paper has previously been circulated under the titles "The impact of COVID-19 employment shocks on suicide and safety net use: An early-stage investigation" and "The impact of COVID-19 employment shocks on suicide and poverty alleviation programs: An early-stage investigation". This work was supported by JSPS KAKENHI Grant Number JP20K01733. All the codes, data and graphs used in the paper are available at https://doi.org/10.7910/DVN/TN7Y1M and https://github.com/michihito-ando, where some estimation results and graphs are presented in more interactive forms. We thank the editor, two anonymous reviewers, and seminar participants for useful comments and Masaya Waki for excellent research assistance.

We are also grateful to Ren Onishi for sharing with us the data of second-tier safety net programs provided by the Ministry of Health, Labour and Welfare.

## Author Contributions

**Conceptualization:** Michihito Ando, Masato Furuichi.

**Data curation:** Michihito Ando, Masato Furuichi.

**Formal analysis:** Michihito Ando.

**Funding acquisition:** Michihito Ando.

**Investigation:** Michihito Ando, Masato Furuichi.

**Methodology:** Michihito Ando.

**Project administration:** Michihito Ando.

**Supervision:** Michihito Ando.

**Validation:** Michihito Ando, Masato Furuichi.

**Visualization:** Michihito Ando.

**Writing – original draft:** Michihito Ando, Masato Furuichi.

**Writing – review & editing:** Michihito Ando, Masato Furuichi.

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
