## [Decision Letter · Decision Letter 0]

13 Jul 2021

PONE-D-21-14904

The impact of COVID-19 employment shocks on suicide and safety net use: An early-stage investigation

PLOS ONE

Dear Dr. Ando,

Thank you for submitting your manuscript to PLOS ONE. After careful consideration, we feel that it has merit but does not fully meet PLOS ONE’s publication criteria as it currently stands. Therefore, we invite you to submit a revised version of the manuscript that addresses the points raised during the review process.

Specifically, as mentioned by Reviewers 1 and 2, the interpretation that the effect of COVID-19 on suicide rate is solely driven by its effect on unemployment is not justified as there can be other channels. As identifying all the possible channels might not be feasible, you should revise the text by acknowledging the possibility of other mechanisms, some of which are discussed by the reviewers. You mention in page 6 that the employment shock is a "major, though not exclusive, causal pathway" but this is not enough. Acknowledging this in the abstract,  intro and conclusion is necessary. 

Reviewer 1 discusses identification and in particular the definition of treatment and control states. It is important to clarify that your identification is based on the differential exposure at the level of prefecture. Your design exploits the fact that some prefectures were exposed more to employment shocks than others assuming that they followed similar trends in employment in the time period before Covid. You can clarify this in the text around equation (1).

Reviewer 1 raised some concerns about the structure of the paper. I agree that you could improve the structure relabelling section 2 as Data and Descriptive Statistics. In this section you can start by mentioning your data sources,  present the trends of Figure 1 and also move the data exploration that is now in Section 3.4. The data exploration can motivate nicely the research design that you discuss in Section 3. In Section 3, I would probably start with equation (2), followed by the definition of the exposure variable (EmpShock) and then by the different outcome variables. Since you are working with prefecture-level data I do not see why you would need to cluster your standard errors (see comment by R1) but I guess you are computing robust standard errors.

Reviewer 2 asks for more information on the institutional background related to safety net programs that you should provide in the restructured Section 2. You may also devote a subsection to institutional background if deemed necessary.

Reviewer 1 is raising some concerns about data limitations which you should mention in the conclusions as limitations of the study. This is also related to the estimates you obtain for some of the safety net outcomes in which there are many cases of pretends. These estimates are not robust and not very convincing so you should discuss these limitations clearly in the text. Alternatively, due to data limitations you may consider focusing only on suicide rates and unemployment benefit as the two main outcomes.

Finally, you should revise the statements in the paper that attempt to provide external validity of your estimates in other settings outside Japan.

We look forward to receiving your revised manuscript.

Kind regards,

Konstantinos Tatsiramos

Academic Editor

PLOS ONE

Reviewers' comments:

Reviewer's Responses to Questions

**Comments to the Author**

1. Is the manuscript technically sound, and do the data support the conclusions?

Reviewer #1: Partly

Reviewer #2: Yes

2. Has the statistical analysis been performed appropriately and rigorously? 

Reviewer #1: No

Reviewer #2: Yes

3. Have the authors made all data underlying the findings in their manuscript fully available?

Reviewer #1: No

Reviewer #2: No

4. Is the manuscript presented in an intelligible fashion and written in standard English?

Reviewer #1: Yes

Reviewer #2: Yes

5. Review Comments to the Author

Reviewer #1: This paper is linked to two streams of literature. One the one hand, it contributes to

the new debate on the consequences of Covid-19 on socio-economic outcomes. On the

other hand, it deals with a consolidated literature about the effect of the unemployment

on suicide and poverty/safety net use. In my view, the paper presents some issues that

need to be addressed. For example, the arguments to support the causal relation between

employment shock and suicide are not entirely convincing; the identication strategy is

not fully clear and the data utilized in the analysis are incomplete. I will discuss these

and other specic issues in the attached report. Finally, the paper is hard to be read

in some of its parts. The structure of the work itself should be revised and results have

to be presented with more transparency.

Reviewer #2: Referee Report for: The impact of COVID-19 employment shocks on suicide and safety net use: An early-stage investigation

Manuscript Number: PONE-D-21-14904

Summary:

The paper presents one of the first sound evidences of the impacts of COVID-19 on suicide rates and safety net use for the context of Japan. The authors claim that what they identify are the impacts of the different incidence of COVID-19 induced employment loses at the regional level on suicide and safety net utilization. Their analysis is developed at the prefecture level in Japan, using a difference-in-difference strategy comparing the values for 2020 with respect to 2019. They present their results in event-study type of models, which allow the reader to check the parallel trend assumption as well as to understand the timing and dynamics of the effects.

Their results show important effects on the two types of outcomes analyzed. More specifically, a one percentage-point increase in the regional unemployment rate translates into an additional 0.39 suicides, which implies an increase by 30% with respect to the pre-pandemic level. The authors also find that these effects are stronger for females than for males.

Similarly, they also report significant effects on some of the safety net outcomes such as the Temporary Loan Program, the Housing Security Benefit and the Public Assistance Program. They do not find significant results for unemployment benefits but they provide a reliable argument, as an important proportion of individuals that are in that job loss situation are not entitled to the unemployment benefit scheme.

The paper provides sound estimates of a very important element of the pandemic that will become relevant in many countries in the coming months. Also, it represents a clear contribution to the existing literature that has been unable to provide similar sound methods and results for those types of outcomes in the context of the COVID-19 pandemic.

In what follows, I enumerate some points that, I believe, could help improving the current version of the paper.

1. My main comment refers to the strong argument that the authors try to make to convince the reader that their findings are solely driven by differential COVID-19 induced changes in the unemployment rate across prefectures in Japan. There could be many other elements caused by the pandemic that might be directly affecting the mental health and suicidal attempts of the population that are unrelated to unemployment probabilities. Therefore, although I understand that distinguishing the impact of all the COVID-19 driven channels is impossible (as many of them are happening at the same time within a prefecture), I would encourage the authors to lower down the intensity of their argument and to acknowledge (in several parts of the paper) that their results might be, partly, driven by other non-unemployment related effects of the COVID-19 pandemic.

2. Following my first comment above, in page 6 the authors write that “Japan’s infection rate was among the lowest in developed countries and major social distancing measures had been implemented at the national level rather than the regional level”. However, it has been documented for several countries that the EFFECTIVE reduction in mobility (and, therefore, the percentage of people isolating themselves at home) is different across regions (even if social distancing measures are dictated at the national level) and depend on several factors such as the employment composition, the age and education level of individuals, etc…

3. Furthermore, in the next paragraph in page 7 the authors state that “Japan’s first declaration of a state of emergency was announced on April 7 for seven high-risk prefectures”. Therefore, there is (indeed) also variation at the regional level on the implementation of the State of Alarm across prefectures (both when introduced as well as when lifted, as the authors acknowledge in the paper).

4. Thus, for all the reasons above, I believe that the authors have to be softer when attributing their effects solely on the differential changes of unemployment across prefectures and have to be much more careful in the interpretation of their results as well as in the abstract, introduction and conclusion sections by acknowledging that other channels might be simultaneously at play.

5. In page 7 when describing the unemployment benefits in Japan, the authors mention that “the coverage rate of the unemployment benefits among unemployed was less than 30% in Japan in the early 2010s”. Could the authors update this coverage rate for a more recent year as, for example, 2019?

6. When discussing the safety net programs analyzed in the paper (in section 3.2), can the authors provide some information on the replacement rate of each of the programs? That is, the amount that an individual gets in each of these programs and the characteristics that determine eligibility. This is important to understand the extent to which individuals in each of these programs are able to maintain the same standard of living than when employed which, in turn, is an important element of mental health status.

7. In figure 4, in plots (e) and (f) there is a sharp drop in recipients in 2020-01 that only recovers the pre-pandemic level in 2020-04; do the authors have a plausible explanation for this evolution that does not occur for the other safety net outcomes? Was it more difficult to apply to these programs in the first phase of the pandemic because of the mobility restrictions and the State of Alarm?

6. PLOS authors have the option to publish the peer review history of their article (what does this mean?). If published, this will include your full peer review and any attached files.

Reviewer #1: No

Reviewer #2: No

---

## [Author Response · Author response to Decision Letter 0]

12 Nov 2021

Please read the attached reply letter.

---

## [Decision Letter · Decision Letter 1]

14 Dec 2021

PONE-D-21-14904R1The impact of COVID-19 employment shocks on suicide and safety net use: An early-stage investigationPLOS ONE

Dear Dr. Ando,

Thank you for submitting the revised manuscript to PLOS ONE, which addresses most of the issues raised by the reviewers. There is still one remaining issue that needs to be addressed and therefore we invite you to submit a revised version that addresses this point. In your revision you need to remove any claims of providing a causal effect of COVID-19 on suicide rates. Reviewer 1 has identified well the parts of the paper that need to be revised, including the title which includes the word "impact". An alternative is to replace "impact" with "association". During the revision you may also update the numbers related to the back-of-the-envelop calculations.

We look forward to receiving your revised manuscript.

Kind regards,

Konstantinos Tatsiramos

Academic Editor

PLOS ONE

Journal Requirements:

Reviewers' comments:

Reviewer's Responses to Questions

**Comments to the Author**

1. If the authors have adequately addressed your comments raised in a previous round of review and you feel that this manuscript is now acceptable for publication, you may indicate that here to bypass the “Comments to the Author” section, enter your conflict of interest statement in the “Confidential to Editor” section, and submit your "Accept" recommendation.

Reviewer #1: All comments have been addressed

2. Is the manuscript technically sound, and do the data support the conclusions?

Reviewer #1: Partly

3. Has the statistical analysis been performed appropriately and rigorously? 

Reviewer #1: Yes

4. Have the authors made all data underlying the findings in their manuscript fully available?

Reviewer #1: Yes

5. Is the manuscript presented in an intelligible fashion and written in standard English?

Reviewer #1: Yes

6. Review Comments to the Author

Reviewer #1: I have read the revised version of the paper and the detailed responses to my report. I think that the authors have taken my comments very seriously and tried to address them properly. In my view, despite the efforts done by the authors, the paper is not able yet to establish a clear causal link between COVID-19 induced unemployment shocks and both suicide rates and safety net use. I think that this mostly depends on data limitations (regional data are not suitable for such causal analysis ) and the intrinsic difficulty of the research issue (Covid-19 pandemic involved too many aspects of life potentially correlated with mental health). That said, I think that the paper offers nice descriptive evidence and it is carefully executed. For these reasons, my recommendation is to recognize limitations more explicitly and from the very beginning of the paper. For instance, I would remove “impact” from the title since it implies a causal analysis. One possibility might be to change the title to "COVID-19 employment shocks, suicide and safety net use in Japan: An early-stage investigation." Moreover, I would recognize the descriptive nature of the study in the abstract, in the introduction, and the conclusions more clearly.

7. PLOS authors have the option to publish the peer review history of their article (what does this mean?). If published, this will include your full peer review and any attached files.

Reviewer #1: **Yes: **Vincenzo Carrieri

---

## [Author Response · Author response to Decision Letter 1]

24 Jan 2022

Please read the attached reply letter.

---

## [Editor Report · Decision Letter 2]

18 Feb 2022

The association of COVID-19 employment shocks with suicide and safety net use: An early-stage investigation

PONE-D-21-14904R2

Dear Dr. Ando,

We’re pleased to inform you that your manuscript has been judged scientifically suitable for publication and will be formally accepted for publication once it meets all outstanding technical requirements.

Kind regards,

Konstantinos Tatsiramos

Academic Editor

PLOS ONE

---

## [Editor Report · Acceptance letter]

16 Mar 2022

PONE-D-21-14904R2 

The association of COVID-19 employment shocks with suicide and safety net use: An early-stage investigation 

Dear Dr. Ando:

I'm pleased to inform you that your manuscript has been deemed suitable for publication in PLOS ONE. Congratulations! Your manuscript is now with our production department. 

Kind regards, 

on behalf of

Prof. Konstantinos Tatsiramos 

Academic Editor

PLOS ONE